# Impact of Hearing Aids on Poverty, Quality of Life and Mental Health in Guatemala: Results of a before and after Study

**DOI:** 10.3390/ijerph17103470

**Published:** 2020-05-15

**Authors:** Mark Spreckley, David Macleod, Brenda González Trampe, Andrew Smith, Hannah Kuper

**Affiliations:** 1Health and Social Care/Allied Health Sciences, London South Bank University, London SE11 0AA, UK; markjamess@hotmail.co.uk; 2Department of Infectious Disease Epidemiology, London School of Hygiene & Tropical Medicine, London WC1E 7HT, UK; david.macleod@lshtm.ac.uk; 3Independent Consultant, Guatemala City, Guatemala; traduccionesfg@gmail.com; 4International Centre for Evidence in Disability, London School of Hygiene & Tropical Medicine, London WC1E 7HT, UK; Andrew.smith@lshtm.ac.uk

**Keywords:** Guatemala, poverty, quality of life, mental health, hearing aids

## Abstract

There are 466 million people globally with disabling hearing loss, many of whom can benefit from hearing aids. The aim of the study was to assess the impact of providing hearing aids on poverty, mental health, quality of life, and activities, among adults in Guatemala. A nonrandomised before and after study was conducted, with a comparison group to assess for secular trends. Adult cases with bilateral hearing impairment were identified within 150 km of Guatemala City, as well as age- and sex-matched comparison subjects without disabling hearing loss. All participants were interviewed with a semistructured questionnaire, and cases were offered hearing aids. Participants were reinterviewed 6–9 months later. We interviewed 135 cases and 89 comparison subjects at baseline and follow-up. At baseline, cases were poorer than comparison subjects with respect to individual income (*p* = 0.01), household income (*p* = 0.02), and per capita expenditure (PCE) (*p* = 0.003). After provision of hearing aids, median household income improved among cases (*p* = 0.03). In the comparison group, median individual income (*p* = 0.01) and PCE (*p* = 0.03) fell between baseline at follow-up. At follow-up, there were also improvements in productive time use, quality of life, and depressive symptoms among cases, but these were less apparent in the comparison group. In conclusion, this study has demonstrated a positive effect of hearing aids in improving quality of life, economic circumstances and mental health among Guatemalan adults.

## 1. Introduction

Hearing loss is very common; an estimated 466 million people globally live with disabling hearing loss, defined as >40 dBHL average at frequencies 0.5, 1, 2, and 4 kHz in the better hearing ear in adults (15 years or older) or >30 dBHL in the better hearing ear in children (0 to 14 years) [1]. This figure equates to a global prevalence of 6.1%, and includes 34 million children and one in three people aged 65 years and older. Overall, approximately 90% of those affected by disabling hearing loss live in low and middle income countries (LMICs), yet services are limited in those settings [2].

Disabling hearing loss has wide-ranging impacts. It diminishes the capacity to detect and localise sounds and recognize speech, and thereby adversely affects communication. As a consequence, people with hearing loss may find it difficult to participate in different activities, particularly those that require social interaction and communication [3,4]. For instance, people with hearing loss on average have attended formal schooling for fewer years and achieved lower educational outcomes than people with normal hearing [5]. Hearing loss is also related to poorer quality of life [6,7] and poorer mental health [8,9]. People with hearing loss may be more vulnerable to dementia [10,11], perhaps because they receive less auditory stimulation. On average, they also experience more difficulties accessing healthcare services [12], and have a higher risk of mortality [13]. There are also economic implications, as hearing loss is linked to lower employment [5], including in highly skilled jobs [14], likely because of difficulties in communicating but also because of the earlier exclusion from education experienced by people with hearing loss. These exclusions can result in a substantial economic impact of hearing impairment at the national level. Estimates from the USA suggest that in 2005 the economic cost due to lost earnings is above $100 billion annually for the 24 million hearing-impaired individuals who do not use hearing aids [15]. WHO estimates that the annual cost of unaddressed hearing loss in 2015 was $750–790 billion globally [16]. The impacts of hearing loss may vary depending on the age at onset, but little evidence is available on this point, particularly from LMICs.

Many people with disabling hearing loss can benefit from provision of hearing aids. A hearing aid does not restore normal hearing or repair the underlying damage that has caused the hearing loss. However, the aid can help to improve sound detection and speech understanding, and thereby improve a person’s ability to take part in everyday life. Evidence from high-income countries shows that provision of hearing aids has benefits [17], including in terms of improving health-related quality of life [17,18], and potentially on cognition [19]. However, the quality of the available data is of concern, and data are lacking for LMICs, especially for multidimensional outcomes.

The aim of this study was to assess the impact of hearing aid provision on multidimensional outcomes, including poverty, mental health, quality of life and activity participation, among adults living in Guatemala. Guatemala is a lower middle-income country in Central America. A nonrandomised before and after study with a comparison group was conducted to assess the impact of hearing aids provision on poverty, quality of life, depression, functional activity and participation, among adults with disabling hearing loss compared to comparison subjects without disabling loss. The study participants were selected from adults living in Guatemala, and the follow-up period was 6–9 months.

## 2. Materials and Methods

### 2.1. Ethics Approval and Consent to Participate

Ethical approval for the study was granted from the ethics committees of the London School of Hygiene and Tropical Medicine and the Zugueme Comite Etica Independiente, based in Guatemala City. All research participants gave written informed consent to take part in the study. People identified as having hearing loss were referred to services for diagnosis and, where appropriate, provision of hearing aids.

### 2.2. Participants

We aimed to identify 200 cases with disabling hearing loss and 200 comparison subjects without disabling hearing loss. This sample size was appropriate to detect a 30% improvement in quality of life after provision of hearing aids, with 80% power and 95% confidence (taking into account loss to follow-up and hearing aid noncompliance, so that 100 cases who were using hearing aids were available at baseline and follow-up).

We identified adult cases who had disabling hearing loss through the community outreach, ear-health screening activities of the Sonrisas que Escuchan Foundation (http://sonrisasqueescuchan.org.gt/), based in Guatemala City in 2015. An experienced audiologist screened potential cases through pure-tone audiology, using calibrated, portable equipment. Cases were eligible for inclusion if they were aged ≥15 years, their bilateral hearing loss was classified as “disabling” (measured as 41 dB and above in the better ear), and they lived within 150 km of Guatemala City. Furthermore, cases were only included if they were judged to be eligible for subsidized hearing aid provision, as they were unable to finance the service independently, and so would be from relatively poor households.

We asked each case to identify three neighbors or nonrelated people living in their community of a similar age (+/− 5 years) and of the same sex to identify a matched comparison group. All potential comparison subjects were visited in their homes and underwent an auditory screening test by a trained fieldworker using a portable, electronic-tablet-based ‘’Shoe-Box Application’’ audiometer. Participants who presented with disabling hearing loss were excluded from the comparison group and offered a referral and follow-up assessment at the Sonrisas que Escuchan Foundation. Initially, eligible comparison subjects were restricted to those with normal hearing, but after difficulties identifying eligible subjects, the eligibility criteria for comparison subjects were extended so that those with mild hearing loss (26–40 db) were included.

### 2.3. Baseline Data Collection

All cases and comparison subjects were interviewed at baseline (October 2015–January 2016), with a semistructured questionnaire, including the following items:

Poverty was measured through assessing assets, income and per capita expenditure (PCE), based on previous approaches [20] and developed from the World Bank’s Living Standards Measurement Survey, which has been used globally for several decades [21]. Items were included on asset ownership, as well as household characteristics, such as the structure of the walls and roof, type of toilet and water facilities, fuel and utility sources and land ownership. Three questions were included related to personal income and total household income (weekly or monthly) as well as other income sources (e.g., pension, a secondary job, or financial support). To measure PCE, a list of 70 items was included, covering food, beverages, clothing, household utility bills, taxes, education and healthcare costs. For each item, we asked participants about the quantity and value of the product or services used, and whether they were purchased, gifted as payment in kind or home produced. For each item, a financial value in local currency (Guatemalan Quetzal—GTQ) was assigned, which was converted into US$. When we measured expenditure, we used a recall period of one week for items that are bought frequently, but used a time-frame of one month for items that are bought less often. These items were summed to calculate total household expenditure, as well as PCE.

Activities data was collected using an activity list [20], based upon the questionnaire in the World Bank’s Living Standards Measurement Survey [21]. We asked participants if they had been involved in a list of common daily activities during the last week. If they responded positively, we asked if they had been involved in that activity yesterday. The participant then estimated the amount of time that they had spent on each activity in minutes and hours. Activities included household tasks, leisure activities household work and employment.

We assessed depressive symptoms using the Patient Health Questionnaire [22]. This tool consists of nine items on depression symptoms, which are summed to generate a symptom score. It is a validated, self-reporting screening tool for assessing mental health. It has been validated in other central American settings, including Mexico, which neighbours Guatemala [23].

Quality of life was measured using the WHOQOL-BREF, a person-centered tool for assessing subjective well-being, which has been translated into many languages and has good psychometric properties [24]. It includes twenty-six questions divided into four domains, including physical health, psychological health, social relationships and the environment.

Sociodemographic data was also collected, including marital status, education level and literacy level (self-defined as ability to read “not at all”, “little”, or “well”).

A pair of trained fieldworkers administered the questionnaire in Spanish, in the participant’s home, using a mobile data collection tool (the “KoBo Toolbox”).

### 2.4. Hearing Aid Provision and Follow-Up

The Sonrisas que Escuchan Foundation contacted all cases after baseline data collection, and cases were given a clinic appointment for fitting with hearing aids. World Wide Hearing donated behind-the-ear devices, which were brand-new and made by Phonak (Baseo Q15 Model). These instruments are considered to be reliable and quality devices, which provide effective sound quality. They are suitable for people with mild–profound hearing loss.

All cases and comparison subjects were revisited in their homes and reinterviewed in July–August 2016 (6–9 months later) using the structured questionnaire, with some additional components (e.g., hearing aid satisfaction). We measured actual hearing aid usage data, which was electronically downloaded from the device during a follow-up clinic appointment, as well as self-reported hearing aid use from the interview data.

### 2.5. Statistical Analysis

At baseline, we compared the demographic characteristics and outcome variables between cases and comparison subjects using logistic regression, adjusting for the matching variables of age and sex. Our hypothesis was that we would observe an improvement in outcomes for cases between baseline and follow-up, but no change for comparison subjects. We did not adjust for other variables, such as SES, as these may have been on the causal pathway between hearing loss and the outcomes (e.g., depression).

We reported the median income and expenditure at baseline and follow up. A comparison between baseline and follow-up was performed within the cases using a Wilcoxon sign rank test, to account for the paired data. A similar test was performed within the comparison subjects to identify if the secular trends could explain the changes, or lack of changes, observed in the cases. A similar approach was applied to the depression, but Mcnemar’s test was performed as the outcome—depression—was a binary variable. A paired *t*-test was used to compare the mean quality of life scores before and after the intervention. These analyses were repeated for comparison subjects, to assess whether there had been secular trends in these variables during the follow-up time period that needed to be taken into account in interpretation of data. A difference-of-difference comparison was not undertaken, as the study was not adequately powered for this test.

## 3. Results

We identified 201 cases with moderate-profound hearing loss, of whom 21 (10%) were excluded as they had unilateral hearing loss. The remaining 180 cases had bilateral, disabling hearing impairment, including 49% with moderate impairment, 39% severe impairment and 12% profound hearing loss. We identified 263 potential comparison subjects, of whom 120 (46%) were excluded because they had disabling hearing loss (98%) or an inconclusive test result (2%), leaving 143 in the comparison group for inclusion in the study. After follow-up, we reinterviewed 135 cases (75%), with the remainder lost to follow-up (96%) or who had lost or suffered theft of their hearing aid (4%). Among the comparison group, 89 (62%) were reinterviewed, and the remainder were lost to follow-up (89%) or refused to be reinterviewed (11%). Data were restricted to cases and comparison subjects who completed both baseline and follow-up questionnaires.

Cases were more likely to be followed-up than comparison subjects (*p* = 0.01), as were older people (*p* = 0.003), but there was no difference in follow-up by gender or category of PCE (as proxy for socio-economic status—SES). Most (71%) of the cases reported that they used their hearing aids every day, of whom 93% stated that they used them at least four hours each day, and 69% for 8–16 h daily. Self-report data were very similar to actual hearing aid usage data, which showed that 98% of cases wore their hearing aids for at least 1–4 h per day, and 53% wore them for 8–16 h each day.

The cases were somewhat older than comparison subjects, as 66% of the cases were aged 60 years or above, compared to 37% in the comparison group (Table 1). Besides age, cases and comparison subjects were well matched on gender, and had similar profiles in terms of marital status, literacy, educational levels and asset ownership. More people in the comparison group (63%) were in paid work, compared to only 45% of cases, although this difference was not statistically significant.

At baseline, amongst those who were receiving income, individual income was significantly lower among the cases than the comparison subjects (*p* = 0.01), and case households had lower annual household income than comparison households (*p* = 0.02) (Table 2). Similarly, mean PCE at baseline was lower among cases than the comparison group (*p* = 0.003). After provision of hearing aids, there was an improvement in median household income among cases (*p* = 0.03). In the comparison group, there was a fall in median individual income (*p* = 0.01) and PCE (*p* = 0.03) during this time period. Allocation of PCE (e.g., on health, education and leisure) did not change after follow-up among cases or comparison subjects (data not shown).

At baseline, cases spent more time on household tasks than people in the comparison group (Table 3). After hearing aid provision, cases spent less time on household tasks (*p* < 0.001) and more time on paid tasks or self-employment (*p* = 0.05). No changes in time allocation were observed in the comparison group after follow-up.

At baseline, few participants reached the diagnostic threshold for depression, whether they were cases (4%) or in the comparison group (1%) (Table 4). We also did not observe any clear trends in severity of depressive symptoms by case status. However, cases were two times as likely to report any depressive symptoms, as compared to the comparison group (*p* = 0.05). At follow-up, there were significant improvements in depression (*p* = 0.03), depressive symptoms (*p* = 0.02) and severity of depressive symptoms (*p* < 0.01), among cases compared with baseline. Similarly, in the comparison group there were improvements in depressive symptoms (*p* = 0.01) and severity of depressive symptoms (*p* = 0.03).

At baseline, cases had poorer quality of life than comparison subjects, across all domains excepting psychological quality of life and overall quality of life, where the differences were not statistically significant (Table 5). After hearing aid provision, cases reported significant improvements in all domains of quality of life, except social relationships. By contrast, the comparison group experienced improvements only in the psychological and environmental domains, and a significant decline in the social relationships domain.

## 4. Discussion

We found that at baseline, adults with disabling hearing loss living around Guatemala City were poorer than the comparison group without disabling hearing loss in terms of individual and household income and PCE. Furthermore, cases spent more time on household tasks, were more likely to experience depressive symptoms and had poorer quality of life than the comparison group. After receipt of hearing aids, cases spent more time on paid work or self-employment, and experienced improvements in household income. At the same time, the comparison group suffered a fall in individual income and PCE, as well as in the social relationships score, which was not apparent in the cases. These data are consistent with a worsening in economic conditions in the comparison group, with provision of hearing aids buffering cases from these impacts. Both cases and comparison subjects reported improvements in mental health at follow-up. Cases showed more substantial gains in quality of life than the comparison group during follow-up, and these were most pronounced in the domains of psychological, social relationships and environmental, and so are plausibly attributable to improved ability to communicate.

These findings are consistent with the existing literature, in terms of the broad impact of hearing impairment. A systematic review investigating the association between poverty and disability in LMICs included 150 studies and showed strong evidence for this relationship, including for people with sensory impairment [25]. Data from high-income settings also confirms the adverse links between hearing loss and employment and earnings [5,14]. Previous studies have also demonstrated a high prevalence of mental health conditions in people with hearing loss [8,9], although evidence is predominantly from high-income countries. One study from Nigeria found that older people experiencing hearing loss showed poorer scores for their activities of daily living and functionality, especially within the emotional domain, which related to depression [26].

The findings also support the existing literature in terms of finding a positive impact of hearing aids, although almost all existing data are from high-income settings. A systematic review of 16 studies concluded that hearing aids improve adults’ health-related quality of life by reducing psychological, social and emotional effects of sensorineural hearing loss [18]. A second systematic review included five randomised controlled trials from high-resource countries, and indicated that among adults with mild/moderate hearing loss hearing aids can improve general health-related quality of life (e.g., physical, social, emotional and mental well-being) [17]. The review also showed that hearing aids facilitated people with participation in day-to-day situations, as well as listening to others. The effectiveness of hearing aids at improving cognition or mental health is less clear in the existing literature from high-income settings [19,27,28].

The literature from LMICs is far sparser, and includes mostly small-scale impact studies. However, these have also shown that hearing aid usage contributes towards better quality of life and mental health. For instance, one study from Turkey with a relatively small sample size (including only 37 participants) demonstrated that depressive signs reduced and psychological state and mental function improved within 3 months of hearing aid use in an elderly population [29]. Similarly, another study with a small sample size (n = 50) among an elderly population in Brazil showed that provision of hearing aids resulted in better self-assessed quality of life, and this was apparent across functional, emotional, social and mental-health domains [30]. More evidence is available, however, of the impact of other interventions in LMICs on people with functional impairments. For instance, the Cataract Impact Study showed that improving visual acuity through provision of cataract surgery improved levels of poverty, activities and quality of life among older people in Kenya, the Philippines and Bangladesh, and that this improvement was sustained over at least 6 years [20].

There are a number of strengths and limitations that must be taken into account when interpreting the findings of the study. Unexpectedly, we detected mild–severe hearing loss among 66% of potential comparison participants. Consequently, we extended the eligible age range from 5 to 10 years in order to reach the necessary sample size of the comparison group. However, this meant that on average cases were older than comparison subjects, with the increased possibility of residual confounding by age. Additionally, the eligibility criteria for comparison subjects was extended so that those with mild hearing loss (26–40 db) were included. This change could mean that cases and comparison group would be more similar in key outcomes, so that as a result the study could have been under-powered to detect differences between the two groups. The study population consisted of adults living within a 150 km radius of Guatemala City, making generalisability to other settings uncertain. The cases were selected from a low socio-economic status group, who could not afford hearing aids and so were eligible for subsidized hearing aid provision, and this may have over-estimated differences in poverty at baseline between cases and comparison groups, although comparison subjects were selected from the same communities. A drop in the median income for the comparison subjects was observed between baseline and follow-up, making it more complex to make inferences on the impact of hearing aids. However, population-based economic data were not available for this time period in this region of Guatemala, making it difficult to understand the fall in the income in the comparison group, and the implications for the study findings. Follow-up was also relatively short, at just 6–9 months. In terms of strengths, multidimensional outcomes were measured using validated tools, and a comparison group was included to adjust for secular trends.

## 5. Conclusions

This study has filled an important evidence gap by demonstrating a positive effect of hearing aids in improving quality of life and mental health among Guatemalan adults. There is also evidence that hearing aid provision may have buffered cases from the economic downturn experienced by comparison subjects during the follow-up period. It is well known that there are currently large gaps in coverage and accessibility of hearing aids, despite their potential positive impacts, as the international production and supply of hearing aids meets less than 10% of the global need [2]. This situation is even worse in LMICs, as fewer than 3% of those who need hearing devices have access. This gap is largely because of the lack of financing and the prioritization of ear and hearing care, and consequently a lack of availability of audiologists or other relevant healthcare workers, and a lack of hearing aids. As a consequence, although the benefits of hearing aid provision are potentially vast and diverse, they are often not realized due to lack of services. The outcomes of this research can be used to support advocacy efforts to call for further scale-up of hearing screening and provision of hearing aids.

## Figures and Tables

**Table 1 ijerph-17-03470-t001:** Baseline sociodemographic characteristics of cases and comparison subjects.

Baseline Characteristics		Cases	Comparison Subjects	Age–Sex Adjusted OR (95% CI)	Age–Sex Adjusted *p*-Value
**Age**	<40	15 (11%)	14 (16%)	Reference	<0.01
	40–49	10 (7%)	11 (12%)	0.87 (0.28, 2.69)	
	50–59	21 (16%)	31 (35%)	0.62 (0.25, 1.56)	
	60–69	43 (32%)	23 (26%)	1.69 (0.69, 4.12)	
	70+	46 (34%)	10 (11%)	4.14 (1.52, 11.29)	
**Gender**	Male	75 (56%)	39 (44%)	Reference	0.18
	Female	60 (44%)	50 (56%)	0.68 (0.38, 1.20)	
**Marital Status**	Single	26 (19%)	20 (22%)	Reference	0.16
	Married/Living Together	85 (63%)	55 (62%)	0.91 (0.42, 1.98)	
	Divorced/Separated	5 (4%)	8 (9%)	0.60 (0.16, 1.70)	
	Widowed	19 (14%)	6 (7%)	2.44 (0.82, 7.23)	
**Literacy**	Not at all	8 (6%)	3 (3%)	1.89 (0.49, 7.36)	0.37
	Little	13 (10%)	5 (6%)	1.85 (0.64, 5.39)	
	Well	114 (84%)	81 (91%)	Reference	
**Education Level**	No Education	10 (7%)	4 (4%)	1.59 (0.47, 5.35)	0.57
	Primary	45 (33%)	34 (38%)	0.84 (0.48, 1.49)	
	Secondary/University	80 (59%)	51 (57%)	Reference	
**Asset Score**	Quartile 1 (poorest)	36 (27%)	20 (22%)	Reference	
	Quartile 2	41 (30%)	19 (21%)	1.18 (0.52, 2.68)	0.14
	Quartile 3	31 (23%)	23 (26%)	0.64 (0.28, 1.47)	
	Quartile 4 (least poor)	27 (20%)	27 (30%)	0.50 (0.22, 1.12)	
**Employed**	Yes	61 (45%)	56 (63%)	Reference	0.09
	No	74 (55%)	33 (37%)	0.59 (0.33–1.08)	

**Table 2 ijerph-17-03470-t002:** Income and per-capita expenditure of cases and comparison subjects, at baseline and follow-up.

	Baseline			Follow-Up			
	Cases	Comparison Subjects	Age and Sex-Adjusted *p*-Value Comparing Cases and Comparison Subjects at Baseline	Cases	Comparison Subjects	*p*-Value for Change in Cases (Baseline Versus Follow-Up)	*p*-Value for Change in Comparison Subjects (Baseline Versus Follow-Up)
Median individual income US$ (IQR)	155 (0, 405)	271 (88, 445)	0.01	121 (0, 405)	162 (0, 472)	0.25	0.01
Median household income US$ (IQR)	490 (277, 828)	614 (324, 1147)	0.02	506 (234, 1012)	540 (292, 1080)	0.03	0.70
Mean PCE US$ (SD)	99 (89)	202 (381)	0.003	111 (147)	124 (167)	0.21	0.03

**Table 3 ijerph-17-03470-t003:** Mean percentage time spent in different activities for cases and comparison subjects, at baseline and follow-up.

	Baseline			Follow-Up			
	Cases	Comparison Subjects	Age and Sex-Adjusted *p*-Value Comparing Cases and Comparison Subjects at Baseline	Cases	Comparison Subjects	*p*-Value for Change in Cases (Baseline Versus Follow-Up)	*p*-Value for Change in Comparison Subjects (Baseline Versus Follow-Up)
Household Tasks	39%	33%	0.01	28%	33%	<0.001	0.73
Paid/Self Employment	9%	11%	0.53	16%	14%	0.05	0.33
Household Work	6%	4%	0.63	5%	5%	0.81	0.49
Social Visits	12%	13%	0.74	14%	8%	0.41	0.18
Leisure Activities	28%	32%	0.07	25%	33%	0.85	0.92
Daytime Sleeping	4%	3%	0.69	7%	3%	0.12	0.61
Other	3%	4%	0.61	5%	2%	0.23	0.22

**Table 4 ijerph-17-03470-t004:** Depression symptoms of cases and comparison subjects, at baseline and follow-up.

		Baseline			Follow-Up			
		Cases	Comparison Subjects	Age and Sex-Adjusted *p*-Value Comparing Cases and Comparison Subjects at Baseline	Cases	Comparison Subjects	*p*-Value for Change in Cases (Baseline Versus Follow-Up)	*p*-Value for Change in Comparison Subjects (Baseline Versus Follow-Up)
Depression	No	130	88	0.50	135	89	0.03	0.32
	Yes	5	1		0	0		
Depressive symptoms	No	97	78	0.05	112	85	0.02	0.01
	Yes (≥minimal)	38	11		23	4		
Severity of depressive symptoms	Not	97	78	0.20	112	85	<0.01	0.03
	Minimal	19	7		18	3		
	Minor	13	4		4	0		
	Moderate->Severe	6	0		1	1		

**Table 5 ijerph-17-03470-t005:** Quality of life of cases and comparison subjects, at baseline and follow-up.

	Baseline			Follow-Up			
Mean (SD)	Cases	Comparison Subjects	Age and Sex-Adjusted *p*-Value Comparing Cases and Comparison Subjects at Baseline	Cases	Comparison Subjects	*p*-Value for Change in Cases (Baseline Versus Follow-Up)	*p*-Value for Change in Comparison Subjects (Baseline Versus Follow-Up)
Overall Quality of Life	3.6 (0.9)	3.8 (0.7)	0.20	3.8 (0.9)	3.9 (0.7)	0.01	0.56
Overall Health	3.3 (1.0)	3.7 (0.8)	0.004	3.7 (0.8)	3.7 (0.8)	<0.01	0.65
Physical	14.6 (2.3)	15.8 (2.1)	0.003	15.0 (2.5)	16.2 (2.1)	0.01	0.07
Psychological	14.6 (2.1)	15.1 (2.0)	0.08	15.5 (1.9)	16.0 (1.9)	<0.01	<0.01
Social Relationships	15.3 (2.2)	15.9 (1.7)	0.005	15.6 (1.8)	15.3 (1.8)	0.22	0.01
Environmental	12.8 (1.8)	13.4 (1.6)	0.001	13.7 (1.7)	13.8 (1.9)	<0.01	0.02

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
