# Peer review of "Impact of Hearing Aids on Poverty, Quality of Life and Mental Health in Guatemala: Results of a before and after Study"

_ijerph, 2020, doi:10.3390/ijerph17103470_

Round 1

Reviewer 1 Report

Overall, the manuscript is well written and presents an interesting dataset from a country that doesn’t get reported on much. My main concern is about the “so what” question. Why wouldn’t one expect outcomes to improve after hearing aids have been provided? How does this study help allocation of funding in a climate where funding is scarce? Please also see my more detailed comments below:

  • Abstract: Please explain the study design more clearly. Is this really a non-randomised controlled study? It sounds more like a case-control study. Or a pre-test post-test design with a demographically matched comparison group. It’s just that a non-randomised controlled study design makes the assumption that the control group would also potentially benefit from the intervention, which allows you to estimate the size of the intervention effect by comparison across groups. However, in your case, the control group doesn’t have hearing problems, and therefore this doesn’t apply here. Perhaps the word comparison group is more appropriat here.
  • Abstract: Is “interviewed” the right term here? Replace with “surveyed”? “Interviewed sounds like a qualitative study, but then you present p-values, so that doesn’t match.
  • Abstract: Did the outcomes for the cases improve to such an extent that they are now equivalent to those of the comparison group?
  • Last sentence of Abstract: The conclusion should focus on something this study found. Rather than adverse effects (this study only looked at baseline levels to inform this statement, but did not explicitly explore whether hearing loss actually decreased QOL etc – this could be a sampling error), the conclusion should focus on the finding that provision of hearing aid improves outcomes.
  • Introduction: The first paragraph nicely sets the scene and illustrates that people in LMICs are proportionally more affected by hearing loss then people in OECD countries. You mentioned that “approximately 90% of those affected by disabling hearing loss live in low and middle income countries (LMICs)”. This statement only alludes to the dramatic skew towards prevalence in LMICs but lacks sufficient detail unless information is provided about the percentage of people living in LMICs.
  • Introduction: Given the multi-dimensional nature of the impact of hearing loss (that the authors themselves acknowledge), the introduction is too brief and only glosses over some of the relevant material. Particularly the dynamics between socio-economic status (SES), health status, and employment in LMICs aren’t sufficiently outlined. Here, also the distinction needs to be drawn between congential hearing loss (possibly much less affected by differences in SES) and accidental hearing loss. Occupational hearing loss will likely be much higher in lower SES groups, and this may be the reason for the relationship between hearing loss and income (a relationship that is likely to be stronger for lower SES – hence expectation for an interaction). Hearing loss due to age may also affect lower SES individuals more as they may need to work past 65 years. These types of complexities and interactions need to be outlined more. Obviously, some of these are better raised in the Discussion section, but some of these issues may also be relevant here already as awareness of them would have informed the design of of the study, choice of measures etc.
  • Given the focus on Guatemala and the previous discussion of LMICs, it would be helpful to place Guatemala on this spectrum here.
  • Line 42: “to range in different activities”? Change to “to participate in different activities”?
  • The section presented in 2.1. is something that most readers would expect to find at the end of the Introduction rather than beginning of Methods section.
  • In the Methods section, the authors mention that participants were “eligible for subsidized hearing aid provision”. Does this suggest that the participants were from the lower and possibly medium SES? Please comment and discuss where relevant.
  • Section 2.4.: It is good to see detailed information about PCE. However, what would also be useful to information on reliability and validity of that measure.
  • Section 2.4.: The section is a bit unbalanced, with a lot of description about the PCE, and others, such as the Patient Health Questionnaire only getting a one-sentence paragraph. Again – readers will need to know whether these measures can be assumed to have sufficient reliability and validity for the particular population they were used for in this study.
  • What was the attrition rate? Did 100% of the participants at baseline also participate in the follow-up interviews?
  • Section 2.6.: Provide more detail here. The nonparamatric Wilcoxon test would be suitable when data are non-normally distributed. Was this the case here?
  • Table 1: The different use of the word baseline is confusing here. For example, why does the word “baseline” appear in the line <40? It sounds like this group formed a baseline. And what do the numbers in that column mean? The table should be comprehensible without having to go back and forth to the text.
  • Table 2: This drop in median income for the controls is dramatic. The authors allude to an economic downturn. This needs to be commented on much more. It complicates the analyses and interpretation of results a bit (the authors later talk about buffering) – nothing one can do about, bit still need to discuss.
  • Table 3: If, among the cases, the percentage of people in employment only increased from 9 to 16% (which may also explain the decrease in household tasks), then the effects of the hearing aid on lifestyle was fairly minimal overall (still meaningful for that small proportion – but for most people things didn’t change so much). So, what was driving the QOL effect? It doesn’t seem to be via new employment opportunities. That’s where an item analysis as opposed to domain level might be quite insightful. Are there a few items that experienced dramatic change and thus increased the domain means?
  • There is the danger that this study is just stating the obvious. Why wouldn’t one expect QOL to improve after hearing aids have been provided? How does this study help prioritise health interventions in a climate where funding is scarce?

Author Response

Overall, the manuscript is well written and presents an interesting dataset from a country that doesn’t get reported on much. My main concern is about the “so what” question. Why wouldn’t one expect outcomes to improve after hearing aids have been provided? How does this study help allocation of funding in a climate where funding is scarce? Please also see my more detailed comments below:

    • Abstract: Please explain the study design more clearly. Is this really a non-randomised controlled study? It sounds more like a case-control study. Or a pre-test post-test design with a demographically matched comparison group. It’s just that a non-randomised controlled study design makes the assumption that the control group would also potentially benefit from the intervention, which allows you to estimate the size of the intervention effect by comparison across groups. However, in your case, the control group doesn’t have hearing problems, and therefore this doesn’t apply here. Perhaps the word comparison group is more appropriat here.Response: We have clarified that it is a before and after study with a comparison group, and use comparison group (rather than control) throughout for clarity. 
    •  
    •  
    • Abstract: Is “interviewed” the right term here? Replace with “surveyed”? “Interviewed sounds like a qualitative study, but then you present p-values, so that doesn’t match.Response: We have clarified the sentence to read “interviewed using a semi-structured questionnaire”
    •  
    •  
    • Abstract: Did the outcomes for the cases improve to such an extent that they are now equivalent to those of the comparison group?Response: We did not conduct formal analyses to test this question, but generally the levels were still lower in cases than the comparison group. We would prefer not to do these additional analyses as it was not part of our analysis plan and our priority is to compare the change in cases before and after intervention.
    •  
    •  
    • Last sentence of Abstract: The conclusion should focus on something this study found. Rather than adverse effects (this study only looked at baseline levels to inform this statement, but did not explicitly explore whether hearing loss actually decreased QOL etc – this could be a sampling error), the conclusion should focus on the finding that provision of hearing aid improves outcomes.Response: The last sentence has been revised to read “In conclusion, this study has demonstrated a positive effect of hearing aids in improving quality of life and mental health among Guatemalan adults.”
    •  
    •  
    • Introduction: The first paragraph nicely sets the scene and illustrates that people in LMICs are proportionally more affected by hearing loss then people in OECD countries. You mentioned that “approximately 90% of those affected by disabling hearing loss live in low and middle income countries (LMICs)”. This statement only alludes to the dramatic skew towards prevalence in LMICs but lacks sufficient detail unless information is provided about the percentage of people living in LMICs.Response: We have adjusted this sentence to emphasise that the majority of people with hearing loss are in LMICs, yet services are limited in those regions.
    •  
    • Introduction: Given the multi-dimensional nature of the impact of hearing loss (that the authors themselves acknowledge), the introduction is too brief and only glosses over some of the relevant material. Particularly the dynamics between socio-economic status (SES), health status, and employment in LMICs aren’t sufficiently outlined. Here, also the distinction needs to be drawn between congential hearing loss (possibly much less affected by differences in SES) and accidental hearing loss. Occupational hearing loss will likely be much higher in lower SES groups, and this may be the reason for the relationship between hearing loss and income (a relationship that is likely to be stronger for lower SES – hence expectation for an interaction). Hearing loss due to age may also affect lower SES individuals more as they may need to work past 65 years. These types of complexities and interactions need to be outlined more. Obviously, some of these are better raised in the Discussion section, but some of these issues may also be relevant here already as awareness of them would have informed the design of of the study, choice of measures etc. 
    • Response: More information has been provided in the introduction outlining the impacts of hearing loss, and specifying that these may vary by age at onset.
    • Given the focus on Guatemala and the previous discussion of LMICs, it would be helpful to place Guatemala on this spectrum here.Response: We have now specified that Guatemala is a lower middle-income country.
    •  
    •  
    • Line 42: “to range in different activities”? Change to “to participate in different activities”?Response: This change has been made.
    •  
    •  
    • The section presented in 2.1. is something that most readers would expect to find at the end of the Introduction rather than beginning of Methods section.Response: This change has been made. 
    •  
    •  
    • In the Methods section, the authors mention that participants were “eligible for subsidized hearing aid provision”. Does this suggest that the participants were from the lower and possibly medium SES? Please comment and discuss where relevant.Response: Yes – this is correct, they are likely to have been relatively poor. This is now clarified in the methods (participant section) and the limitations section of the discussion.
    •  
    •  
    • Section 2.4.: It is good to see detailed information about PCE. However, what would also be useful to information on reliability and validity of that measure.Response: The measure of PCE was based on the World Bank’s Living Standards Measurement Study, which has been used globally for the last decades. This is now clarified in the methods.
    •  
    •  
    • Section 2.4.: The section is a bit unbalanced, with a lot of description about the PCE, and others, such as the Patient Health Questionnaire only getting a one-sentence paragraph. Again – readers will need to know whether these measures can be assumed to have sufficient reliability and validity for the particular population they were used for in this study.Response: More information has been provided on the activity measure and on PHQ-9.
    •  
    •  
    • What was the attrition rate? Did 100% of the participants at baseline also participate in the follow-up interviews?Response: The sample was limited to people who were included in both baseline and follow-up interviews. This is now clarified in the first paragraph of the results.
    •  
    •  
    • Section 2.6.: Provide more detail here. The nonparamatric Wilcoxon test would be suitable when data are non-normally distributed. Was this the case here?
  • Response: Yes – this is correct and has now been clarified in the methods.

  • Table 1: The different use of the word baseline is confusing here. For example, why does the word “baseline” appear in the line <40? It sounds like this group formed a baseline. And what do the numbers in that column mean? The table should be comprehensible without having to go back and forth to the text.Response: This is now worded as “reference” rather than “baseline”, for clarity.
  •  
  •  
  • Table 2: This drop in median income for the controls is dramatic. The authors allude to an economic downturn. This needs to be commented on much more. It complicates the analyses and interpretation of results a bit (the authors later talk about buffering) – nothing one can do about, bit still need to discuss.Response: We have clarified in the limitations section of the discussion that “A drop in the median income for the comparison subjects was observed between baseline and follow-up making it more difficult to make inferences on the impact of hearing aids. However, population based economic data was not available for this time period in this region of Guatemala, making it difficult to understand the fall in the income in the comparison group, and the implications for the study findings 
  •  
  •  
  • Table 3: If, among the cases, the percentage of people in employment only increased from 9 to 16% (which may also explain the decrease in household tasks), then the effects of the hearing aid on lifestyle was fairly minimal overall (still meaningful for that small proportion – but for most people things didn’t change so much). So, what was driving the QOL effect? It doesn’t seem to be via new employment opportunities. That’s where an item analysis as opposed to domain level might be quite insightful. Are there a few items that experienced dramatic change and thus increased the domain means?Response: The improved QoL is mostly within the domains of social relationships and psychological, and so is likely to be a result of improved ability to communicate. This is now clarified in the discussion. “Cases showed more substantial gains in quality of life than the comparison group during follow-up, and these were most pronounced in the domains of psychological, social relationships and environmental, and so are plausible attributable to improved ability to communicate.
  •  
  •  
  • There is the danger that this study is just stating the obvious. Why wouldn’t one expect QOL to improve after hearing aids have been provided? How does this study help prioritise health interventions in a climate where funding is scarce?
  • Response: The findings on QoL are unsurprising, but there has been a lack of data previously on the impact on economic productivity or mental health, which this study now adds. We have strengthened the conclusion further to explain how these findings can support advocacy efforts.

Reviewer 2 Report

This is a straightforward study to explore the impact of hearing aid provision on poverty, participation in a number of activities, quality of life and mental health in Guatemala City, Guatemala. The study involved comparisons before and after hearing aid provision for both cases and controls. I have several methodological or statistical modeling concerns and hope the authors can address them adequately.

First, it was unclear what type of statistical analyses the authors conducted. I would imagine that the authors used repeated measures since both cases and controls were measured at baseline and follow-up. However, no where did the authors provide this information. It was simply stated that they used both logistic and linear regressions for their data analyses pending on how the dependent variables were measured. As such, I am not sure if the authors used the models pertaining to independent samples or dependent samples.

Second, I suggest that the authors use either fixed effects or random effects models for panel data analysis. The key results should be whether the dependent variables were statistically different or improved overtime when the baseline and follow-up measures were compared for cases, whereas for controls no such significant differences would be anticipated. These should be made crystal clear upfront.

Third, it seems that the authors only controlled for age and gender in their multivariate regression analyses. Is this correct? If so, then the findings were not robust. For example, when predicting depressive symptoms via PHQ-9 and the quality of life via the WHOQOL-BREF, the authors should control for additional confounding factors such as educational attainment, employment status, marital status, and poverty status. Without these controls, how could we know if the differences between baseline and follow-up would be mediated away by these confounders? If the findings stay the same after adding these controls, then I am convinced that providing hearing aid indeed improved multiple aspects of respondents’ life, especially income, mental health and the overall quality of life.

Author Response

First, it was unclear what type of statistical analyses the authors conducted. I would imagine that the authors used repeated measures since both cases and controls were measured at baseline and follow-up. However, no where did the authors provide this information. It was simply stated that they used both logistic and linear regressions for their data analyses pending on how the dependent variables were measured. As such, I am not sure if the authors used the models pertaining to independent samples or dependent samples.

Response: We have now given more information on the types of statistical analyses conducted.

“At baseline, we compared the demographic characteristics and outcome variables between cases and comparison subjects using logistic regression, adjusting for the matching variables of age and sex. We did not adjust for other variables, such as SES, as these may be on the causal pathway between hearing loss and the outcomes (e.g. depression).

We compared the change in the repeated measures of outcomes measured at baseline and follow-up for the cases, and separately for the comparison subjects. Parametric tests were used, unless data were non-normally distributed. Therefore, logistic regression analyses were undertaken to compare the follow-up and baseline measures for cases with respect to employment and expenditure. A Mc-Nemar’s test was used to compare the baseline proportions with the follow up proportions of depression. Wilcoxon sign-rank test was used to compare baseline and follow-up median income, asset score and depression. A paired t-test was used to compare the mean quality of life scores before and after the intervention. These analyses were repeated for comparison subjects, to assess whether there had been secular trends in these variables during the follow-up time period which needed to be taken into account in interpretation of data.

Second, I suggest that the authors use either fixed effects or random effects models for panel data analysis. The key results should be whether the dependent variables were statistically different or improved overtime when the baseline and follow-up measures were compared for cases, whereas for controls no such significant differences would be anticipated. These should be made crystal clear upfront.

Response: We concluded that non-parametric methods were more appropriate, given the distribution of the variables (e.g. for income, many people had no income, and there was a long tail). These models could therefore not be used in our analyses. We have clarified that we compared baseline/follow-up changes separately in cases and comparison subjects.

Third, it seems that the authors only controlled for age and gender in their multivariate regression analyses. Is this correct? If so, then the findings were not robust. For example, when predicting depressive symptoms via PHQ-9 and the quality of life via the WHOQOL-BREF, the authors should control for additional confounding factors such as educational attainment, employment status, marital status, and poverty status. Without these controls, how could we know if the differences between baseline and follow-up would be mediated away by these confounders? If the findings stay the same after adding these controls, then I am convinced that providing hearing aid indeed improved multiple aspects of respondents’ life, especially income, mental health and the overall quality of life.

Response: In the comparison of case and comparison subjects we adjusted for the matching variables of age and sex. We did not adjust for other variables, such as SES, as these may be on the causal pathway between hearing loss and depression (as an example). Moreover, the variables the reviewer mentioned (marital status, literacy, education level, asset score), did not vary between cases and comparison subjects as shown in Table 1, and so could not play a strong confounding role. This is now clarified in the text.

Reviewer 3 Report

Please find my comments in the text. I will email it to Maja Volcevic
Assistant Editor, MDPI DOO.

Also, you can use references from the published article in the Journal of Clinical Medicine: The Effect of hearing Aid Use on Cognition in Older Adults: Can We Delay Decline or Even Improve Cognitive Function?

Best Regards,

Author Response

Abstract: Nothing mentioned about the sample size and the methodology employed.

Response: This information is provided in the abstract. I am unclear as to what is missing

Statistical analysis:

Response: More information is now provided.

Table 1 comments:

Response:

  • Age was included as a categorical variable, with no assumption as to the pattern of the relationship of age and hearing impairment. We therefore do not believe it would be helpful to include age squared term, and we did not make assumptions about effect modification by age.
  • Literacy level was self-defined and this is now clarified in the methods.
  • We have corrected the formatting of the Asset Score in the table
  • We would prefer to retain Confidence Intervals, rather than change to standardised coefficients, as these will be more familiar to the reader and therefore easier to interpret.

Discussion

  • We have rephrased “small study” as requested and SES has been written in full.

Conclusions

  • We have rewritten the conclusions to be more informative and to draw out the implications.  

Also, you can use references from the published article in the Journal of Clinical Medicine: The Effect of hearing Aid Use on Cognition in Older Adults: Can We Delay Decline or Even Improve Cognitive Function?

Response: This reference has now been added in the introduction and discussion.

Round 2

Reviewer 1 Report

Thank you for the thorough revisions.

Reviewer 3 Report

The authors supposed to attach their report on how they implemented my previous comments. I am not satisfied with the re-submission.